# Context-Aware Edge-Based AI Models for Wireless Sensor Networks—An Overview

**DOI:** 10.3390/s22155544

**Published:** 2022-07-25

**Authors:** Ahmed A. Al-Saedi, Veselka Boeva, Emiliano Casalicchio, Peter Exner

**Affiliations:** 1Department of Computer Science, Blekinge Institute of Technology, SE-371 79 Karlskrona, Sweden; veselka.boeva@bth.se (V.B.); emiliano.casalicchio@uniroma1.it (E.C.); 2Department of Computer Science, Sapienza University of Rome, 00185 Roma, Italy; 3Sony, R&D Center Europe, SE-221 88 Lund, Sweden; peter.exner@sony.com

**Keywords:** edge computing, artificial intelligence, wireless sensor network, context-awareness

## Abstract

Recent advances in sensor technology are expected to lead to a greater use of wireless sensor networks (WSNs) in industry, logistics, healthcare, etc. On the other hand, advances in artificial intelligence (AI), machine learning (ML), and deep learning (DL) are becoming dominant solutions for processing large amounts of data from edge-synthesized heterogeneous sensors and drawing accurate conclusions with better understanding of the situation. Integration of the two areas WSN and AI has resulted in more accurate measurements, context-aware analysis and prediction useful for smart sensing applications. In this paper, a comprehensive overview of the latest developments in context-aware intelligent systems using sensor technology is provided. In addition, it also discusses the areas in which they are used, related challenges, motivations for adopting AI solutions, focusing on edge computing, i.e., sensor and AI techniques, along with analysis of existing research gaps. Another contribution of this study is the use of a semantic-aware approach to extract survey-relevant subjects. The latter specifically identifies eleven main research topics supported by the articles included in the work. These are analyzed from various angles to answer five main research questions. Finally, potential future research directions are also discussed.

## 1. Introduction

Recent advances in technology have had a great impact on today’s digital world, surrounded by billions of intelligent sensors integrated with the Internet of Things (IoT) [1,2]. In such a complex dynamic environment, IoT devices, which usually have limited computing power and small memory capacity, can constantly generate huge amounts of data that can be analyzed in remote cloud data centers. Wireless Sensor Networks (WSN) are considered to be one of the key technologies used for data generation in IoT components.

However, transferring data from where it is generated to a data center increases communication overhead and bandwidth consumption, and also raises privacy concerns. Thus, the use of cloud processing alone is clearly not the most efficient approach for real time systems (e.g., health monitoring, autonomous driving, smart city) [3]. Thus, it is necessary to conduct the computation of the data collected by sensors as locally as possible, incorporating intelligence from edge devices, to move computation from cloud to edge [4]. This means placing a kind of artificial intelligence close to edge devices capable of processing complex behaviors and adapting to rapidly changing situations. In addition, more than ever before, microcontrollers are powerful enough to make intelligent decisions without any external help based on data collected from various sensors [5]. These devices can also analyze data, transmit data to low-latency actuators, and only transfer summarized information to the cloud [6]. In short, adding some sort of intelligence to the sensor nodes represents the next step in fulfilling an “awareness” level to the edge [7,8,9,10]. If data from different sensors are appropriately combined, the integrated data can be more precise, more reliable or simply provide a better understanding of the context in which the data was obtained [11]. Thus, sensor fusion will continue advancing in almost all applications, including security, logistics, voice recognition, object detection, etc. In such a sensor environment, most of these applications focus on performance metrics such as latency, reliability and even security [12,13,14].

As new wireless technologies such as WiFi Direct, 5G, Zigbee, LoRa, NB-IoT and LiFi are rapidly being developed, Edge Computing (EC) will soon help transition processing and analysis from the cloud to the edge [15,16,17].

### 1.1. Our Contribution

The purpose of this study is to provide an overview and understanding of the theoretical background, challenges, approaches, motivations, and gaps for the implementation of intelligent context awareness in wireless sensor networks. This document presents a literature review that covers the analysis of research documents published in the period between 2015 and January 2022 and available from the Scopus and Web of Science databases. In previous related reviews, we have not found an exhaustive work that explores deeply intelligent solutions using artificial intelligence (AI), machine learning (ML), or deep learning (DL) algorithms and their contributions to building the context-awareness in various WSN applications. Moreover, this study can be distinguished from the existing related surveys through the following key contributions.

Apply a semantic-aware approach to identify survey-relevant subjects.Identify and explore various AI/ML and DL methods that can be used in the establishment of a context awareness setting of sensor networks.Analyze key challenges and the research gaps found in the literature that need to be solved.Discuss the motivations for integration of intelligent context awareness in wireless sensor networks.Outline future research directions.

This investigation will fill the research gaps by comparing the included papers in terms of their challenges, strengths, limitations, motivations, and the way forward in the field. Furthermore, this review can help future researchers in identifying and exploring new perspectives in the field of sensor network context awareness field.

### 1.2. Organization of This Work

The rest of the paper is organized as follows. Section 2 introduces the technological context and discusses other surveys related to the topic of this study. Section 3 explains the methodology of this work. Section 4 analyzes the data extracted from included papers and discusses the results related to the defined research questions. Section 5 provides a discussion about industrial perspectives, context-aware challenges and corresponding intelligent solutions in a logistics use case. Finally, we conclude the paper with some open issues in Section 6. Figure 1 exhibits the organization of this paper.

## 2. Background and Related Work

### 2.1. Background

This section introduces the technological context necessary to facilitate the understanding of the context-aware AI modeling challenges in edge environments. Firstly, in Section 2.1.1, we describe the main features of context-aware computing as a specific paradigm within the EC environment. Then, in Section 2.1.2, we present the basics of EC. Section 2.1.3 briefly discusses main concepts of AL, ML and DL. Finally, in Section 2.1.4, we focus on the role of sensors and present examples of applications that use sensors in dynamic environments.

#### 2.1.1. Context Awareness

In recent years, context-aware applications have captured a lot of attention as they extract user context, such as location, activity, time, health status, physical environmental state, etc. Various types of special sensors are used. These can be physical sensors, such as the Global Positioning System (GPS) sensor and accelerometer, or virtual sensors, such as user calendar, weather web service and weather radar [18,19]. However, in a consensus definition, context awareness is defined as “systems that adjust according to conditions: environmental (e.g., the level of pollution), physical (e.g., one’s current location), social (e.g., one’s family and colleagues), or temporal (e.g., the time of the day), as well as changes in these things over time” [20,21]. As part of this article, context is defined as a situation and environment of sensors in WSN. Therefore, contextual information use includes interactions between sensor nodes and the reaction of sensor nodes to environmental changes to discover information of interest [21,22]. A context-aware system architecture is exhibited in Figure 2.

Sensor nodes typically have specific context metrics. Some examples of these metrics are location, energy level, connectivity, speed, temperature, pressure, and link quality.

#### 2.1.2. Edge Computing

EC, a computing paradigm which extends cloud computing, enables all computing outside the cloud to happen at the edge of the network [23], and more specifically in applications where real-time processing of data is required. Lately, the “Edge” defines the point where sensor nodes and IoT devices are located in the local network [24]. EC works on a huge quantity of data generated by sensor nodes or users in the edge network [25]. However, with respect to context awareness, data generated either from a single sensor node or by multiple sensor nodes represent unprocessed data, while context information represents processed raw data [26]. EC system architecture consists of four primary components shown in Figure 3: centralized cloud, a centralized data processing system, operates on a massive amount of data that can be accessed at anytime. Edge data centers, are specialized data centers located closer to the edge than the cloud that deliver faster processing than edge devices, as well as minimal latency and data transmission costs compared to the centralized cloud in real time. Edge devices are pieces of physical hardware that send data between the local network and the central cloud. Traditional edge devices can include many different things, such as edge sensors, routers, firewalls, and chips. Sensor nodes are data accumulation sources. These technologies include edge sensors and chips which are capable of gathering, sensing, and processing data—to an extent.

Three types of EC architectures have been introduced, namely: Mobile-Edge Computing (MEC), fog computing, and cloudlets computing [24]. MEC extends EC by providing compute and storage resources near to low energy, low resource mobile devices. While fog computing seeks to realize a seamless continuum of computing services from the cloud to the things rather than treating the network edges as isolated computing platforms. Cloudlets are small data centers that are typically one hop away from mobile devices [24]. These paradigms differ in terms of software architecture, context awareness, and location of nodes.

#### 2.1.3. AI Disciplines

The definition of Artificial Intelligence (AI) was first coined by McCarthy in the 1950s, where the field of AI refers to the capability of a machine to imitate human intelligence processes [27]. The overall goal of AI research is to let machines perform some advanced decisions that require intelligent humans to complete. The main concern of AI was and still is to do tasks that are typically hard to characterize formally in terms of mathematical rules [28]. The difficulty of explaining this type of task showed that AI approaches needed the ability to find patterns and gain knowledge [28,29]. This ability is defined as ML, which allows computer applications to learn and act on data without explicitly programming it [30]. However, mapping the knowledge gained from learning to final prediction requires the implementation of methods classified as representation learning, in which features are converted into representations including useful information [31]. For complex concepts, if a representation is indicated in terms of other representations, DL needs to be used. DL allows computational models to learn representations with different levels of abstraction. Thus, DL can be seen as representation learning that can imitate human thinking and gain knowledge [28]. These days, AI, ML, and DL are three popular terms that are often used interchangeably to characterize intelligent systems. Their relations are shown in [32], in which, DL is part of ML and is also a part of the broad field of AI while ML is considered a part of the AI umbrella.

#### 2.1.4. Wireless Sensor Network

The WSN, which is the backbone of IoT, consists of dedicated, small, resource-constrained, and low-cost sensor nodes that are randomly deployed in a monitoring environment to perform certain specific tasks over a period of time [33]. Current WSN are widely used in various applications, such as healthcare [34], smart homes [35], environment monitoring [36], etc. Each sensor node has a processing power, radio, and electrical storage device that converts a physical phenomenon of a heterogeneous environment into an electrical signal. The main task is to cooperatively sense, gather and process data about devices in the coverage region, and then transfer it to remote servers for deriving the information [37,38]. Figure 4 displays the typical WSN architecture which contains sensor nodes, fog nodes, and a central cloud.

A number of factors play a role in determining node failures such as harsh environments, restricted energy, and device faults. It is also necessary that the sensor network is able to support the task for a minimum specified period of time [39].

### 2.2. Related Work

With the aim of outlining the contribution of the present study over the existing related surveys, we provide herein a brief overview of these works. There are many surveys on the subject of context-aware computing, context-aware sensor networks and context-aware intelligence related to the subject of our study. Peraraet et al. [40] provide a framework for an overview of context-aware IoT that briefly describes how ML models work, but does not deepen this point. Furthermore, their proposed solutions are yet to be implemented in real time. In [41], a literature review focusing on the most common techniques in the development of context-aware systems is presented. However, they show that all methods have disadvantages and do not dive into a discussion of ML methods. Vahdat et al. [42] expose a survey study about specific application domains, namely Mobile Crowd Sensing (MCS) in smart environments. In [43], authors have aimed to understand the state-of-the-art in the development of context-aware middlewares (CAMs) for aiding the construction of HAR applications when using ML. However, they do not consider explicitly the sensor networks and they have focused only on HAR applications. Sezer et al. [44] focus on Data Analytics in Edge-to-Cloud environments. However, they do not deepen the discussion of ML for HAR. Bogale et al. [45] consider the AI approaches in the context of fog (edge) computing architecture, but the authors do not present a deep discussion of the various ML algorithms that are used. In Preeja and Krishnamoorthy’s study [46], authors have outlined the context characteristics, context organization, and context-aware systems, such as context modeling and the use of a middleware approach to simplify the development considering the heterogeneity of technologies. However, their survey has been developed in the context-aware middleware domain. The work in [47] brings an overview of significant concepts and related applications in various fields of context-aware systems. Although this work has presented a review of the latest development of context-aware systems during the period from 2008 through 2019, the authors present only a few discussions. Chatterjee et al.’s study [48], on the other hand, is the most relevant paper to our review. This paper has focused on identifying the current trends, foundations, and components of the envisioned IoT devices to enable the design of more efficient connected intelligent systems in the future. However, the authors do not deepen in a discussion of the type of AI, ML, or DL solutions that have been used to address the predefined challenges or dive deeper into a discussion of application domains. In [49], authors have focused only on the algorithms and modeling techniques used in context-aware recommenders. The purpose of our study is to fill the gap by considering recent advances in the field of context-aware edge-based AI models for sensor networks and by identifying application domain-independent challenges. Moreover, our work differentiates from the above-mentioned studies by applying a semantic-aware approach for identifying the main subjects supported by the survey included papers. Namely, the articles’ keywords are analyzed by clustering them into groups of semantically similar terms. Thus, in our survey we have managed to extract the major covered topics in its subject framework. Table 1 lists a comparison of our study with the various related reviews conducted in the period between 2015 and January 2022.

## 3. Methodology

Given the latest changes that occurred on advances in edge computing with advances in artificial intelligence, the attention of academia/industries is predominantly focused on the state of the art in context-awareness systems, which are considered crucial for the realization of intelligent IoT and sensor network applications. Therefore, it is necessary to identify the state of the literature on the relationship between AI fields and edge computing to support context-aware sensing systems. To achieve this goal, the authors formulate research questions to define the scope of work.

Leaden by research questions listed in Table 2, this research was carried out by defining the below listed search criteria to gather all relevant publications. We divided this review of the literature into four main phases outlined in Figure 5: data preparation, search conducting, data extraction and analysis, identification of survey topics.

### 3.1. Preparation of the Data

Based on the above research questions (Table 2), this study is focused on the following meanings: “context awareness”, “edge computing”, “artificial intelligence”, “machine learning”, “deep learning”, “sensors” and “wireless sensor network”. In addition to the main concepts, their synonymous were defined. The indexing databases considered for this study were Web of Science and Scopus, which were recommended for conducting a literature review by multiple researches in [50,51]. In line with the setup of the study, a list of inclusion and exclusion criteria were set to improve the selection of publications and to guarantee a successful analysis process. The inclusion criteria are:**IC1.** Journal and conference papers that address the intersection between context-aware, artificial intelligence methods, and sensor network domain, containing the terms in the title, or keywords. Papers with the terms just in the abstract are excluded in this study.**IC2.** Papers available in electronic form published between 2015 and January 2022.**IC3.** Journal and conference papers written in English.

The defined exclusion criteria are:**EC1.** Articles without access to the electronic file.**EC2.** Bibliographic, conference reviews, works of non-indexed or gray literature, and master thesis.**EC3.** Duplicate studies after reading the title.**EC4.** No relevance after reading title and abstract.

Then, the identified studies were sieved according to five defined filters, explained below.

**Filter 1** allows the retention of papers related to context-awareness and AI fields such as ML and DL for sensor networks. The search takes the TITLE + ABSTRACT + KEYWORDS fields as a whole, making those 3 fields into just one and then running a text search (IC1).**Filter 2** allows the retention of publications available in electronic form and published between 2015 and January 2022 (IC2). Articles without access to its electronic file are discarded (EC1).**Filter 3** includes only publications journal and conference papers written in English (IC3). It also allows the removal of bibliographic, conference reviews, works of non-indexed or gray literature, and no research thesis (EC2).**Filter 4** allows the removal of duplicate or redundant publications (EC3).**Filter 5** allows the removal of irrelevant papers. The authors of the current survey have conducted this task by reviewing the title and abstract of each paper and selecting only papers that are related to the topic of the survey (EC4).

### 3.2. Search Conducting

Considering the above-defined research questions, the main focus was papers from the most reputed scientific databases, namely Web of Science and Scopus, during 2015–January 2022. Table 3 presents the search strings of this study.

### 3.3. Data Extraction and Analysis

Initially, a search of scientific papers from the Web of Science and Scopus databases was performed to extract the publications from the selected sources. The selection criteria were divided into the five filters discussed above in order to collect more relevant articles. Therefore, the selection process comprehended the phases depicted in Figure 6 and followed the procedures outlined below:A total of 2760 publications related to context-awareness and AI fields were retrieved, of which, 1841 were obtained from Scopus and 919 from Web of Science.As a result of the second filter, 515 publications from Scopus and 380 publications from Web of Science were retrieved, available in electronic form and published between 2015 and January 2022 with access to its electronic file.Only publications in journals and conference papers written in English were retained. Bibliographic, conference reviews, etc., were excluded. Thus, 435 documents for Scopus and 328 for Web of Science were retrieved as a result of the third filter.After merging the publications of Scopus and Web of Science, 763 duplicate publications were removed in the fourth filter, and 490 left.After reading the title, abstract, and keywords of these publications, 349 were eliminated because they were not related to the topic of the survey. At the end of the fifth filter, 141 papers were left. These are included and examined in this work.

After we managed to extract and classify the data, the aggregated data were then analyzed to be used to respond to the research questions in Section 4.

### 3.4. A Semantic-Aware Approach for Identifying the Survey Main Subjects

We have applied a semantic-aware approach for identifying the main subjects supported by the survey-related articles. The approach is built upon the analysis of the articles’ keywords. Initially, all different (unique) keywords of the extracted articles are gathered together. The number of all unique keywords is equal to 637. Then, this number is reduced to the most frequent keywords. Namely, a score is assigned to each keyword reflecting its frequency of appearing among the articles’ keywords. Then, all keywords which scores are below the preliminary defined threshold (2 in our consideration) are filtered out, i.e., only the most frequent 82 keywords are left. The applied approach uses the semantic similarity between keywords to identify the main research/application subjects covered by the survey. It is based on the idea published in [52]. In order to be able to apply this approach we have manually associated each keyword (from the most frequent ones) with its synonym keywords. This assists us in calculating the semantic similarity between keywords based on the common synonymy between two keywords by using the Jaccard coefficient [53]. Thus, the semantic similarity between two keywords $w_{i}$ and $w_{j}$ can be computed as follows:(1)$$SemSim\left(w_{i},w_{j}\right)=\frac{n_{i}+n_{j}-n_{ij}}{n_{ij}},$$
where $n_{i}$ and $n_{j}$ are the synonymy numbers of $w_{i}$ and $w_{j}$, respectively, and $n_{ij}$ is the synonymy common number between $w_{i}$ and $w_{j}$. The keywords then can be partitioned into groups of semantically similar keywords by applying a selected clustering algorithm (e.g., DBSCAN). The obtained clusters of keywords represent the main research/application subjects supported by the selected articles. The flowchart in Figure 7 illustrates the process of identifying main survey subjects. We have identified 11 of these subjects presented in Figure 8. The 11 keyword groups and the title of their subjects are shown in Table 4. In addition, the relative percentage of cluster size produced by applying DBSCAN with eps = 0.3 on the 82 most frequent keywords are also elaborated in Figure 8. The parameter eps specifies how close data points (keywords) should be to each other to be considered a part of a cluster. We have experimented with different values of eps and 0.3 has produced the most balanced grouping without any outliers.

The selected articles can be further analyzed with respect to identified subjects to obtain deeper insight into the limitations and gaps in the current research related to the survey theme (Q5). For example, each article can be represented by a vector of membership degrees of the article to the different subjects (clusters of keywords). The membership degree of article *i* to subject $S_{j}$ can be calculated as $k_{ij}/n_{i}$, where $k_{ij}$ is the number of keywords from the keyword list of article *i* that belong to cluster $S_{j}$ and $n_{i}$ is the total number of keywords describing article *i*. In this way, a fuzzy distribution of the articles among the identified subjects is obtained.

The fuzzy grouping of the articles can easily be transferred into a non-fuzzy clustering by associating each article to only subject(s), for which it has the highest membership degree. This allows us to evaluate the popularity of each subject quantified by taking into account the number of articles belonging to it (see Figure 9). Each group (research/application subject) can be associated with specific AI/ML techniques and domains of application by further analysis of the challenges and application domains addressed by the articles assigned to it. The knowledge extracted due to this analysis can be used to answer Q2, Q3 and Q5 by facilitating the identification of under/over-represented topics in the current research along with the challenges shared among different application fields. Each cluster of articles can also be studied with respect to the state-of-the-art solutions used to address the issues in the research/application subject presented by this cluster (Q3). The articles can also be grouped with respect to the identified subjects by using their membership degree vectors to measure the similarity between each pair of articles. In comparison with the grouping produced by the first approach where each group of articles is related to one concrete subject, in the current clustering the articles that are grouped together will be similar with respect to more than one research/application subject, e.g., we can identify articles that use the similar AI/ML techniques and at the same time deal with issues in the same application fields. As a result of this grouping the studied articles have been distributed in 15 disjoint clusters. We have experimented with different values for the parameter eps. However, all of those have produced clustering solutions where some of the articles are considered to be outliers. This is due to the scatter of articles in terms of topics, i.e., most of the articles are related to no more than two topics. The value 0.4 for the parameter eps is chosen, since it has produced the less number of outlying articles.

As one can see in Table 5, the top ten keywords that appear the most in the articles included in this review are “ML”, “DL”, “IoT”, “activity recognition”, “sensors”, “HAR”, “wearable sensors”, “CNN”, “classification” and “context-awareness”, i.e., the review perimeter is well outlined by those.

Figure 9 presents the percentage of papers of sample studied per the eleven major subjects identified. The references to the papers related to each subject are given in Table 6. In addition, as it was discussed before the popularity of each identified subject is assessed relatively with respect to the others by taking into account the frequency of the keywords assigned to its cluster. This is represented by a pie chart in Figure 8. Interestingly, the four most popular subjects (see Figure 8) identified based on the keywords’ frequency coincide with the four subjects supported by the quantity of the published papers (see Figure 9). These subjects are AI, ML and DL, Smart Healthcare, Smart and Wearable Devices and Edge Computing and Smart. However, AI, ML and DL has a much higher percentage than Smart Healthcare with respect to the keywords’ frequency while these subjects are equally represented with respect to the published papers. This may be due to the fact that in the case of Figure 9, some papers are cross-disciplinary, i.e., they have the same highest membership degree to more than one subject and in that way, they are counted for all those subjects. One can also notice that Context-Awareness and Energy Consumption and Saving have the same representativeness in both Figure 8 and Figure 9. In addition, the two least popular subjects (Mental Health and Computer Vision) are identical in both figures.

As it was mentioned above, Table 6 exhibits the paper references belonging to each subject and how many belong to these primary subjects. Smart Healthcare stands out with 35 papers. AI, ML and DL and Smart and Wearable Devices are second and third, with 34 and 31 documents, respectively. In the fourth position, it can be found the Sensors and WSN with 17 documents, followed by the Edge Computing and Smart Monitoring with 16 papers. The number of included papers for Computer Vision and Mental Health is equal, i.e., only 5 papers are assigned to each one. It is worth noting that most of these studies, as can be seen in the table, belong to more than one subject, since their keywords are distributed among various clusters of keywords (main subjects).

## 4. Result Analysis

We have analyzed the data extracted by the selected publications (see Section 3.3) to answer each research question presented in Table 2. The research questions are addressed one by one in the following subsections.

### 4.1. Q1: How Much Literature Activity Has There Been between 2015 and January 2022?

We have reviewed the significant research papers in the field published from 2015 to January 2022. Figure 10 presents the details of the year-wise publications (publishing trend). A clear increasing interest in the recent years can be seen from that figure. For each year, we show the total number of papers normalized on monthly base. The highest number of papers published per year are after 2020. This demonstrates not only highly increased interest, but also the high need of research in intelligent context-aware WSNs.

Moreover, the included papers per year are analyzed and distributed in four groups based on the used computational techniques, i.e., ML, DL, ML and DL and AI. These are presented in Figure 11, showing a significant increase in the use of DL in the studies published after 2019 with the expectation that this will continue to flourish in the following years. In addition, one can notice after 2019 the appearance of studies using AI modelling and reasoning techniques such as fuzzy logic.

### 4.2. Q2: What Are the Challenges in Context-Aware Edge-Based AI for Sensor Networks?

In order to answer the question Q2, the main challenges have been identified and are shown in Figure 12. These are Human Activity Recognition (HAR), monitoring, Quality of Service (QoS), energy saving, activity recognition, object detection and location-based service (LBS). They have been addressed by various AI, ML and DL approaches under different application domains as this will be discussed in the answers of the next research question.

According to Figure 12, HAR is the top-addressed challenge, namely in 28.5% of the sample. HAR includes recognizing daily performed locomotion modes [54,55,56,57,153,154], analyzing the behavior of the elderly in daily life [58,134,135], gait analysis [59,155,156,157], etc. Monitoring is the second most studied issue, namely in 26.8% of the included papers. Not surprisingly, in the context of monitoring, various applications have been identified, e.g., health [60,89,115,158,159], smart buildings [90,116,160], agriculture [161,162], stress [61,117,136,163], transportation [91], military defense [164], etc. Other challenges comparatively highly studied in the included papers are QoS [92,93,118,128,137,138,139,152,165,166,167,168,169,170,171,172,173,174] with 14.6%, and energy saving [62,63,94,95,96,114,119,142,143,144,175] with 8.9%.

### 4.3. Q3: What Are the State-of-the-Art Solutions Used to Address the Challenges Depending on the Specific Application Field?

As it was already discussed in the answer of Q2, the two most studied challenges are HAR (28.5%) and monitoring (26.8%), followed by QoS (14.6%), energy saving (8.9%) and activity recognition (8.1%). In addition, as one can notice in Figure 12, LBS is investigated only in 2.4% of the sample. In order to answer Q3, we have initially explored the relationships of these challenges with the application fields addressed in the included papers. The studied application domains are presented in Figure 13.

The top explored category is healthcare studied in 58% of the included papers. It is followed by smart cities (12%), autonomous driving (5%), environment monitoring (5%) and transportation/logistics (4%). All the other categories are below 3%.

It is interesting to study the relationships among the five most studied application domains, the top addressed challenges and used intelligent techniques. In that way, two types of connections will be revealed: one between the state-of-the-art solutions used to address the identified main challenges and the other between the addressed challenges and corresponding application fields used to evaluate the proposed intelligent solutions. This will outline the technological and application perimeter of the context-aware intelligent systems for sensor networks. In addition, Section 5 discusses the identified challenges along with the intelligent techniques used in the logistic use case that has inspired this study.

Figure 14 illustrates the relationship between the HAR challenge and top five application domains. As can be seen in the figures, 66.7% of health care studies have addressed the HAR challenge. Smart cities and transportation are the second and third, with 14.8% and 11.1% of the papers studying this challenge, respectively. HAR is logically less explored in environmental monitoring and autonomous driving applications.

Furthermore, Figure 15 depicts the relationship between QoS with the top five application domains. Healthcare is again the most studied application domain (38.5%), followed by smart cities, transportation, autonomous driving and environmental monitoring sharing the equal interest (15.4%) in the included papers.

In addition, we study the relationships of the identified challenges against AI techniques applied. Figure 16 illustrates the relationship between the HAR challenge and intelligent techniques used to address it in the included papers. According to this figure, 41.4% of the papers studying HAR challenge use traditional ML approaches for handling it, in 44.8% of the papers DL techniques are applied, while in 10.3% of the studies ML and DL approaches are applied together to address this challenge, and only in 3.4% of the papers addressing HAR, AI methods have been employed.

Furthermore, Figure 17 depicts the relationship between the QoS challenge and intelligent methods used to address it. We can observe that in 61.5% of the studies ML techniques have been applied to solve this challenge, in 30.8% of the papers DL methods have been preferred, and AI techniques are only 7.7% of the included papers devoted to QoS.

Figure 18 visualizes, respectively, the relationships between the energy saving challenge and intelligent approaches used to address it in the studied papers. It is worth mentioning that the same trend is observed for the activity recognition challenge.

Figure 19 presents the more frequently used ML/DL algorithms in addressing HAR challenge. We can observe that Decision Tree (DT) (ML) and Convolutional Neural Networks (CNN) (DL) are equally used to address this challenge, namely in 17% of the papers studied the challenge. In addition, 17% of the studies have applied various other approaches such as Linear Regression (LR), active learning, fuzzy logic, etc. Neural Networks (NN) are mentioned only in 13% of approaches, while Random Forest (RF) and Support Vector Machine (SVM) have been implemented in 10% of proposed methods addressing HAR challenge.

Figure 20 depicting specific ML/DL techniques used to address Monitoring challenge shows that in contrast to HAR challenge, a lion’s share (23%) of techniques used is for SVM, followed by 17% for RF and then K-Nearest Neighbor (K-NN) and CNN taking the equal percentage (11%).

Figure 21 depicts the trend donut chart of the ML and DL approaches that have been applied in addressing the Activity Recognition challenge. The results show that two DL techniques (CNN and Recurrent Neural Network (RNN)) and traditional LR are sharing the same percentage of usage, namely 20% of each one, while Deep Neural Network (DNN), K-NN, RF and SVM have also shared the equal usage percentage, but twice lower (10%).

It is worth mentioning that 25% of methods used to address the QoS challenge apply DT while techniques such as CNN, RF, Naive Bayes (NB), K-means and Q-learning have been used only by 12% of the approaches. The analysis of the Energy Saving challenge reveals that 38% of approaches utilized are based on Long Short-Term Memory (LSTM), while each of the techniques K-means, RF, CNN, RNN and DNN has been used in 12% of the studies devoted to this challenge.

With regard to the state-of-the-art solutions used to solve the identified main challenges, we initially analyzed the sample of selected studies from the view of AI. Studies using fuzzy logic techniques are 37.5% of articles, while studies using various other approaches to ML/DL approaches have the attention of 62.5% of articles. In the discussed papers, see Figure 22, the review of the sample studied through the ML lens shows that the most used ML techniques are SVM, RF, DT and K-NN. For example, SVM and RF approaches are used by 17.2% and 14.9% of the selected articles, respectively. DT and K-NN are identified in 12.6% and 10.3% of the studies, respectively. While the clustering techniques are applied only by 5.7% of the selected papers. Regarding other ML techniques used, 28.7% are applied by sample studies.

From the sample studied of DL discipline, illustrated in Figure 23, papers considering CNN are the most numerous, representing 29.3%. NN are discussed in 24.1% of the studies, while LSTM is used by 19.0% of the sample papers. The use of Reinforcement Learning (RL) and DNN is found in 3.4% of the articles.

Figure 24 illustrates the percentage of usage of the ML and DL approaches in the selected studies. The diagram shows that 52% of selected papers have used ML techniques such as SVM, RF, K-NN, clustering, etc., while 39% addressed the edge-based AI challenges by DL approaches such as CNN, NN, LSTM etc. In addition, we have found that 10% of the papers use ML and DL together to address different challenges in context-aware scenarios.

Table 7 presents an overview of selected studies that used ML and DL techniques to address challenges in context-aware scenarios and highlights the techniques used in each of them.

### 4.4. Q4: What Are the Motivations to Adopt AI Solutions to Context Awareness Scenario?

The motivations for applying AI/ML approaches to context-aware scenarios identified in the included studies are shown in Figure 25.

The results show that 44% of the studies have a motivation to improve the recognition. Some of these are proposed to recognize human activity with the wearable devices [54,59,72,82,85,153] or to ease the finding of objects [87,129,151], or to recognize the emotion [98,113]. Another motivation is related to management which refers to the configuration, maintenance, and monitoring (27.2%), such as monitoring the elderly [134,135] or detection of health-related problems [58,69,80,115,158], detection of abnormal driving behavior [64], etc. QoS is also an important motivation with 14.4%, as optimization of resource-constrained IoT devices [137,169,173], or forecast the connectivity and bandwidth of mobile devices [139]. In addition, data privacy and security are another significant motivation, as to address the data privacy concern in healthcare applications [92,165], or in autonomous driving [166], or to improve access control techniques of smart devices [118]. Improving the performance of location-aware applications is the motivation used in 2.4% of the selected studies, e.g., in location tracking applications [101,178] or in autonomous driving technology [109].

### 4.5. Q5: What Are the Limitations of Current Literature or What Are Gaps Existing in the Current Research about Applying AI Technologies to Context Awareness That Future Researchers Can Investigate?

The papers included in the survey are analyzed from three main perspectives: used state-of-the-art (AI, ML, and DL) techniques, application domains, and addressed challenges. The identified limitations and gaps in the study are discussed in light of these three perspectives.

As a result of the analysis conducted in Section 4.3, we have identified the lack of unsupervised and semi-supervised approaches that allow for dealing with the cases of not enough or entirely missing labeled data as well as transfer learning techniques in the reviewed state-of-the-art solutions. These can be considered as a particular gap calling for future research and development of techniques dealing with those challenges typical for most context-aware real-world scenarios. In addition, the current state-of-the-art research in the context-aware intelligent systems is lacking solutions in the framework of collaborative learning where several smart devices share insights from the local training, without sharing the raw data, namely decentralized and distributed learning schemes such as Federated Learning [179] and Swarm Learning [180].

The second perspective that has been studied in the included papers reviews the application domains used to evaluate the proposed intelligent context-aware solutions. Figure 13 exhibits that healthcare domain is studied in more than half of the papers (58%). The percentages of the other identified application domains are very low in comparison to that of healthcare, see the discussion presented in Section 4.3. For example, logistics/transportation domain which is in the focus of our special interest (see Section 5) is studied in only 4% of the reviewed papers. More than the half of the application domains (e.g., smart homes, agriculture, computer vision, Industry 4.0, robotics, sustainability and safety) are even below this percentage, only smart cities show a higher representation, namely the domain is mentioned in 12% of the papers.

Finally, from the perspective of the challenges identified (see Section 4.2), we can observe in Figure 12 that location-based services are understudied, mentioned in only 2.4% of the reviewed articles. The interest in studying energy saving, activity recognition and object detection is also not very high, below 10%, which is quite lower in comparison with HAR and monitoring, each one explored in more than 25% of the studies included.

## 5. Logistics Use Case: Industrial Perspectives, Challenges and Intelligent Techniques

The current survey is inspired by an industrial use case in smart logistics. This and the fact that this application domain is less studied in comparison to the other top four fields motivated us to provide with an additional discussion of the industrial perspectives in logistics, the challenges identified along with the corresponding intelligent solutions used to addressed them.

Logistics is the backbone of global trade with global logistics expenditures making up between 10 to 15 percent of the total world GDP [181]. It is a high volume and low-margin market with many actors, turning supply chains into very complex operations with numerous logistics partners involved in each shipment. Because of this complexity, visibility into where goods are at a given moment, if they have been handled correctly, and if they are going to be delivered on time is a difficult goal to achieve. Most transportations lack visibility and traceability of what happens during the journey, making it difficult to answer when goods will arrive.

With the emergence of IoT trackers, tracking of individual goods has become possible by attaching a tracker on goods instead of manually tracking supply chain segments, such as individual lorries or containers. The decreasing footprint and price point of IoT devices have additionally enabled the ubiquitous deployment to not only high value goods and entire pallets, but to also individual items.

Due to the mobile nature of trackers, they are essentially battery-operated and function in environments where charging is often limited or nonexistent. Trackers often undergo shipment in demanding environments, such as containers and warehouses, in which wireless communication is unattainable and incur a high energy cost. In addition, trackers are often equipped with other sensors, such as temperature and accelerometer, each with their own energy profile. Altogether, trackers need to operate during the entire length of a transportation, while sensing and reporting significant events along a route and maintaining a sufficiently low energy profile.

With the ubiquity of small form factor AI computing and individual goods level tracking, the possibility for trackers to sense their environment and adapt behavior accordingly, both individually and collaboratively together with other devices, has become attainable. As an example, trackers can uncover relations between the device and its operating environment in order to adjust their sensing and operating profiles. For example, detecting indoor/outdoor and providing this context-aware information in various environments may be helpful and lead to battery-saving solutions [182]. In addition, multiple trackers can work in unison to make distributed decisions and utilize sensor sharing for improved power efficiency [183].

The study published in [184] identifies research in the context of intelligent transport logistics as performance enhancing approaches that combine multiple modalities of information technology and sensing into a real-time transportation management system using AI and ML. A central challenge posed by the authors stemming from the black-box nature of AI systems, lies in identifying and finding solutions for mitigating the effects of biased decisions taken by artificial intelligence. Given the advent of continuously learning AI systems wherein decision-making is updated given new input, we believe that this challenge will receive an increasing importance and focus. It is also important to note that the main challenges addressed in the studied papers in logistics domain are HAR, QoS, and monitoring as it is shown in Figure 26. The monitoring challenge is in the focus of 50% of the sampled papers, while each of HAR and QoS is studied in 25% of the logistics devoted papers.

In addition, AI categories identified from the included studies in logistics are shown in Figure 27. Interestingly half of the papers in logistics have used DL techniques, namely CNN, while one-quarter of the papers included have utilized ML and AI equally. This once more confirms the finding identified in [184] concerning the black-box nature of the most existing intelligent solutions in logistics and the need for new more transparent approaches.

## 6. Conclusions and Open Issues

In this paper, we have provided an extensive survey of context-aware edge-based AI methods for WSN technology. Five research questions have been addressed by analyzing 141 research articles used as primary papers. Initially, we have applied a semantic-aware approach for analyzing the keywords of the included papers in order to extract the survey main subjects. Eleven such topics have been identified, e.g., the most popular are AI, ML and DL, Edge Computing and Smart Monitoring, Smart Healthcare and Smart and Wearable Devices. In the analysis carried out, we have also discovered that healthcare, smart cities, autonomous driving, environmental monitoring, and transportation are the top five application domains. Improving the quality of recognition, efficient management, enhancing QoS and efficiency, and ensuring higher security are the top five motivations for enabling intelligent applications in context-aware systems.

Various AI-based solutions have been studied in the included papers. Unsupervised and semi-supervised algorithms, as well as transfer learning techniques are identified as ones that have not grabbed much attention of researchers in most context-aware scenarios. Moreover, other promising collaborative frameworks such as federated learning and swarm learning have not been adequately explored. There is also a lack of research covering the location-based services in the reviewed articles, further studies with more focus on these challenges is highly suggested.

Our future research plans include deeper investigation of the challenges and gaps identified due to the conducted survey in order to expand further the knowledge gained and use those to develop new efficient intelligent edge-based solutions. For example, we are particularly interested in developing decentralized resource-efficient unsupervised or semi-supervised learning frameworks. Our short run goal is the implementation of such a federated framework and its initial study and evaluation in a logistics use case. 

## Figures and Tables

**Figure 1 sensors-22-05544-f001:**
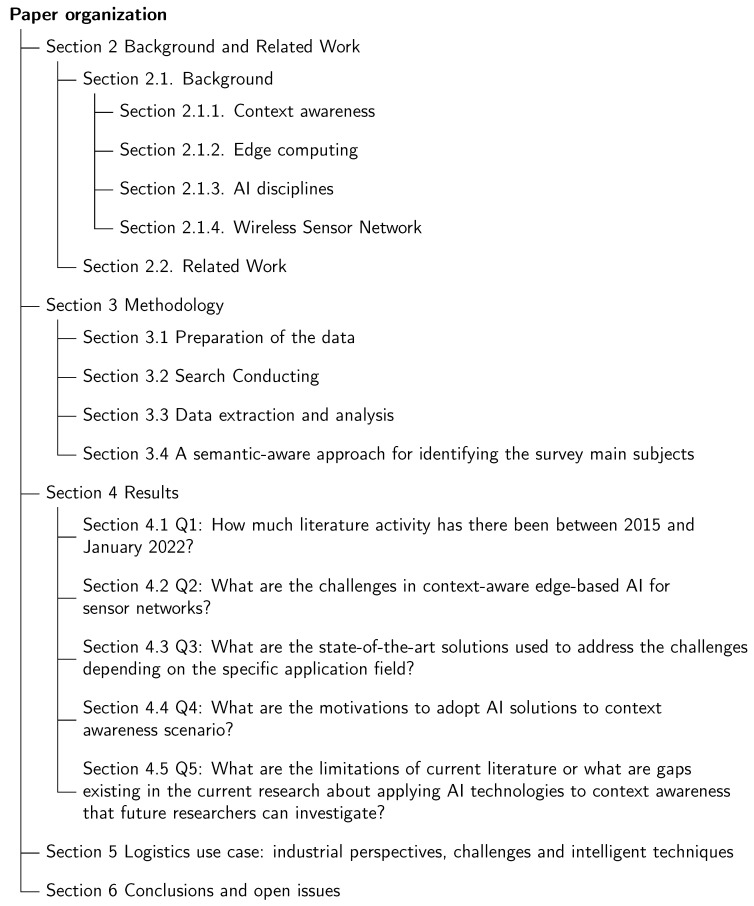
A schematic illustration of the paper organization.

**Figure 2 sensors-22-05544-f002:**
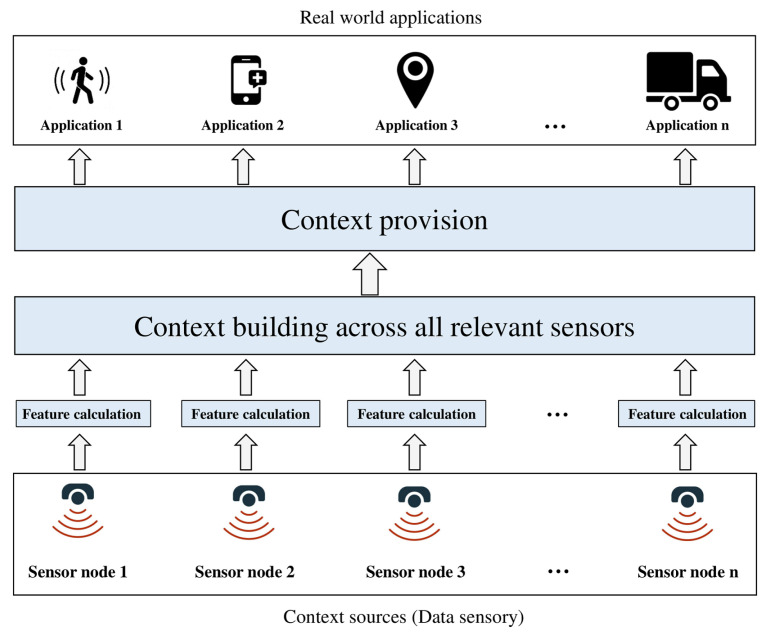
Context-aware framework layers.

**Figure 3 sensors-22-05544-f003:**
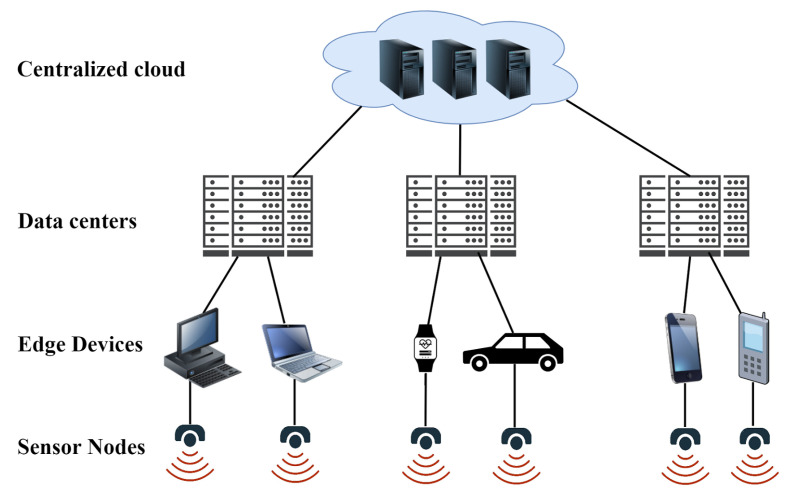
EC ecosystem.

**Figure 4 sensors-22-05544-f004:**
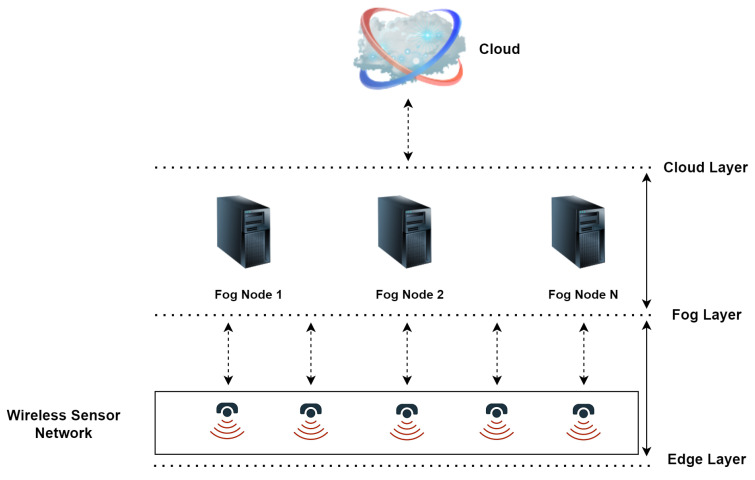
A schematic presentation of the general WSN architecture.

**Figure 5 sensors-22-05544-f005:**
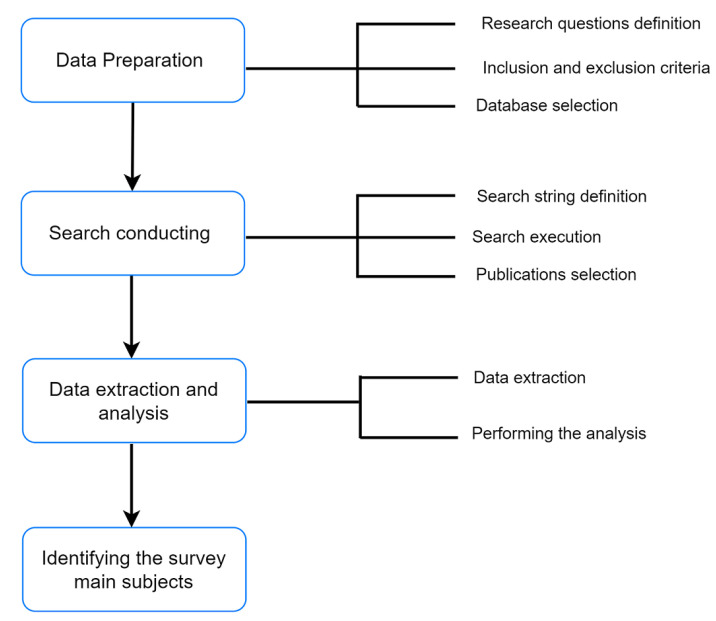
The main methodological phases of the study.

**Figure 6 sensors-22-05544-f006:**
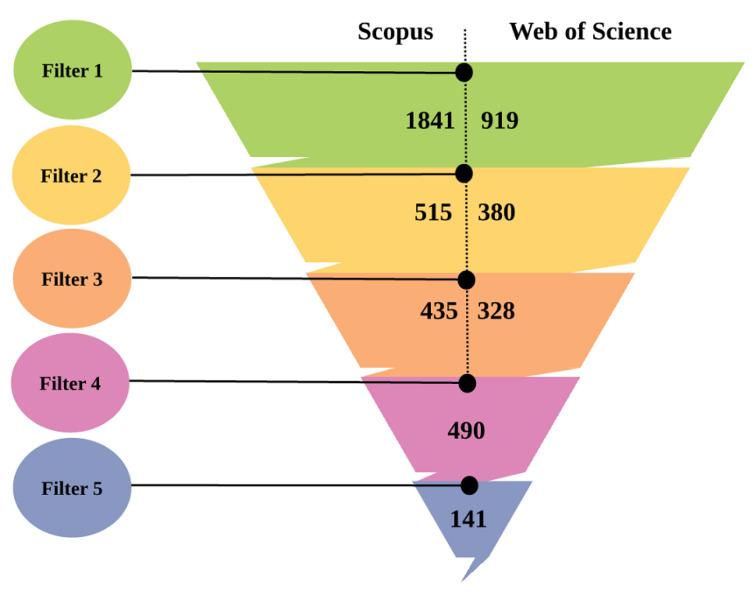
The number of papers selected after applying each filter of the survey’s related papers is given for WoS and Scopus databases, respectively.

**Figure 7 sensors-22-05544-f007:**
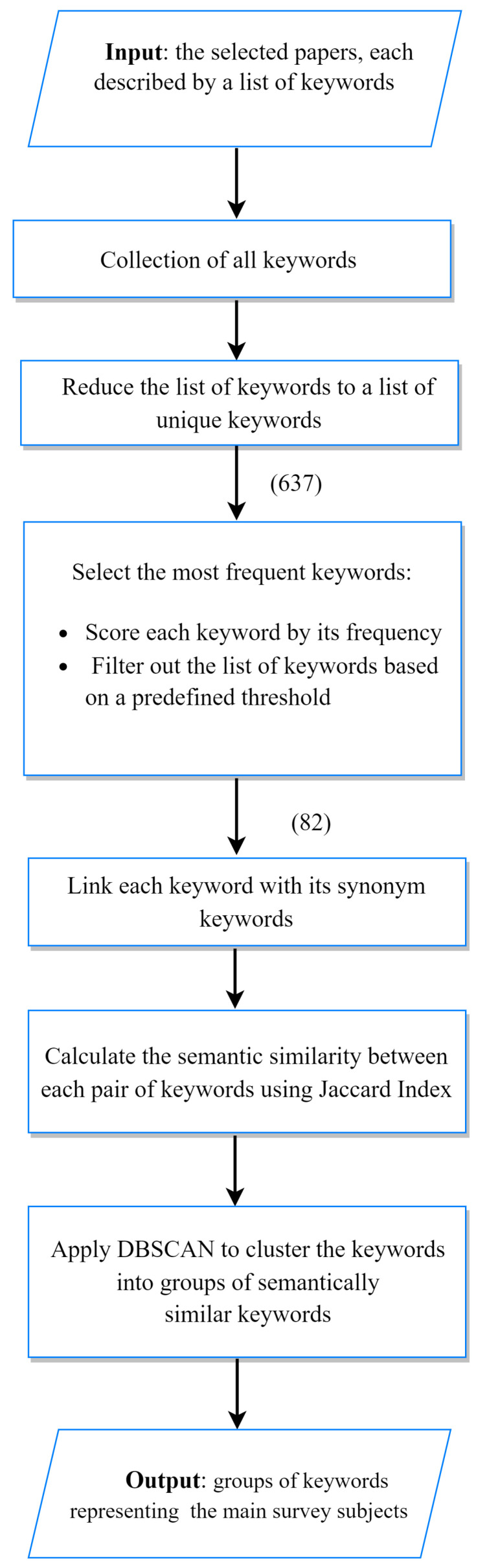
Flowchart describing the different steps of the semantic-aware approach applied to identify the main subjects covered by the included papers.

**Figure 8 sensors-22-05544-f008:**
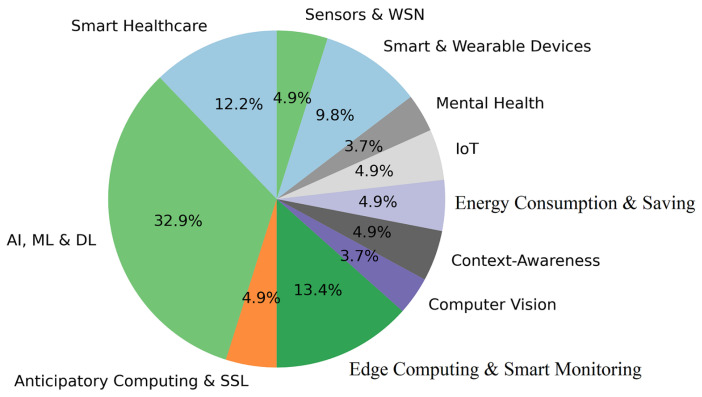
Relative popularity of the identified subjects assessed on the based of the keywords’ frequency. The most popular subject is AI, ML and DL followed by Edge Computing and Smart Monitoring, Smart Healthcare and Smart and Wearable Devices.

**Figure 9 sensors-22-05544-f009:**
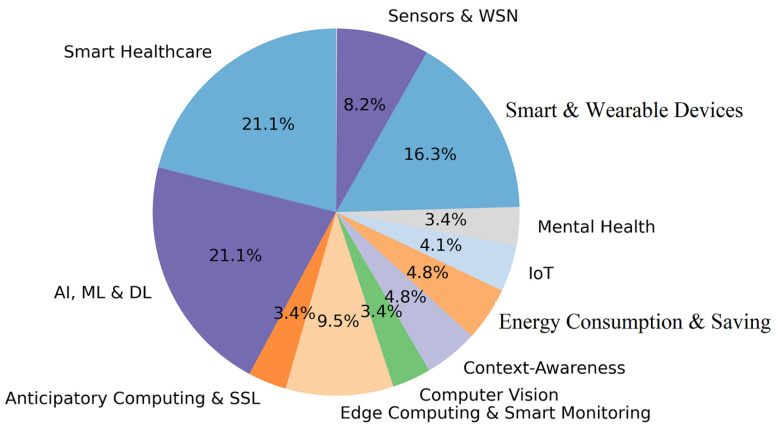
Percentage of papers of sample studied per the main identified subjects. The most represented subjects are AI, ML and DL and Smart Healthcare followed by Smart and Wearable Devices, Edge Computing and Smart Monitoring and Sensors and WSN. These well reflect the survey theme.

**Figure 10 sensors-22-05544-f010:**
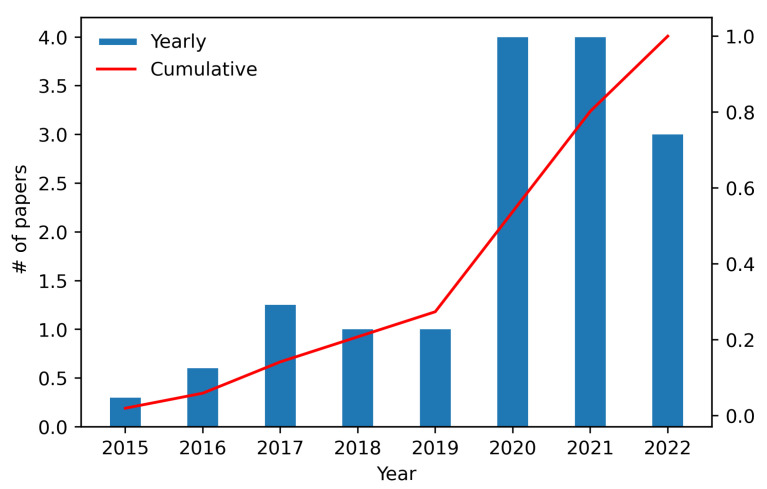
Included papers per year (publishing trend) normalized on monthly base. There was a significant increase in the number of included papers published after 2019.

**Figure 11 sensors-22-05544-f011:**
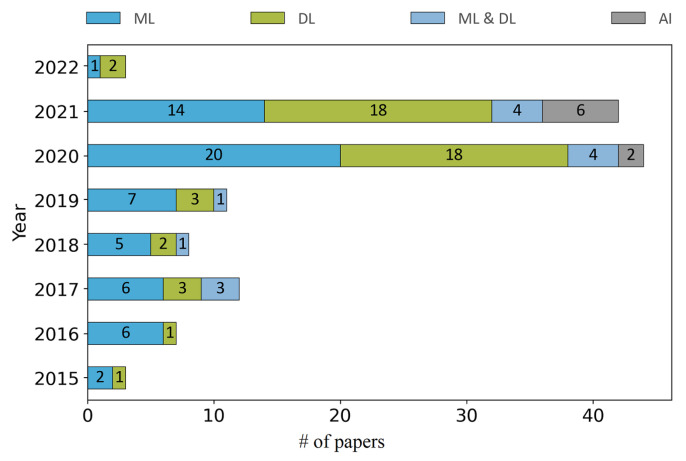
Included papers per year are distributed in four categories based on the used computational techniques, i.e., ML, DL, ML and DL and AI.

**Figure 12 sensors-22-05544-f012:**
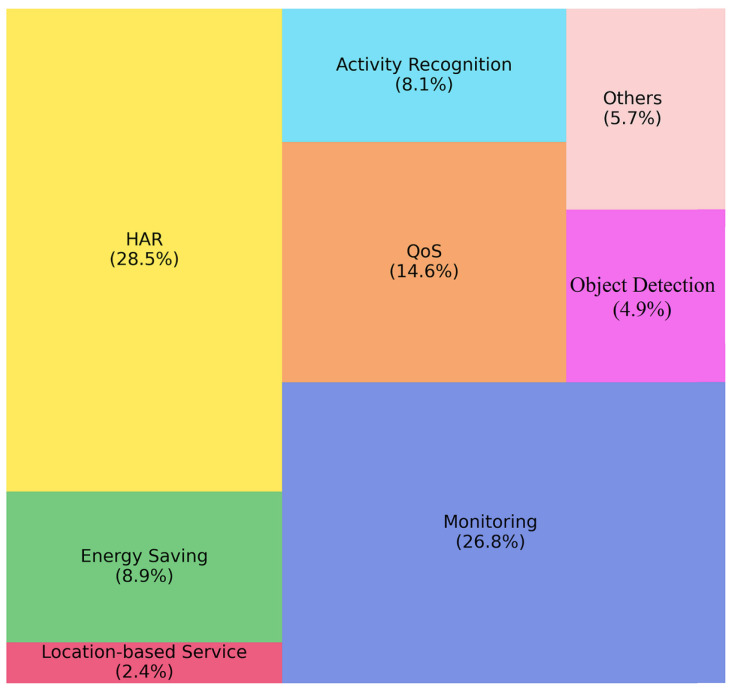
Main challenges addressed by the papers included in the survey.

**Figure 13 sensors-22-05544-f013:**
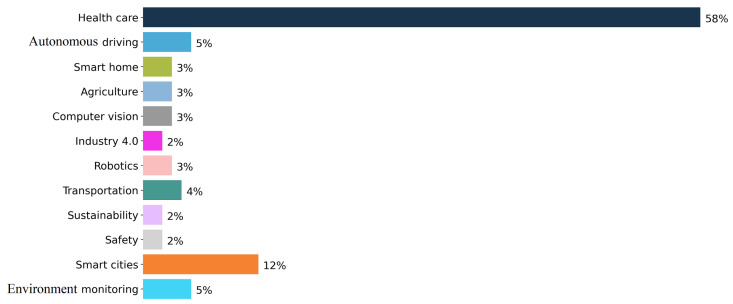
Percentage of papers of sample studies per domain of applications. The most popular category is healthcare followed by smart cities, autonomous driving, environment monitoring and transportation (logistics).

**Figure 14 sensors-22-05544-f014:**
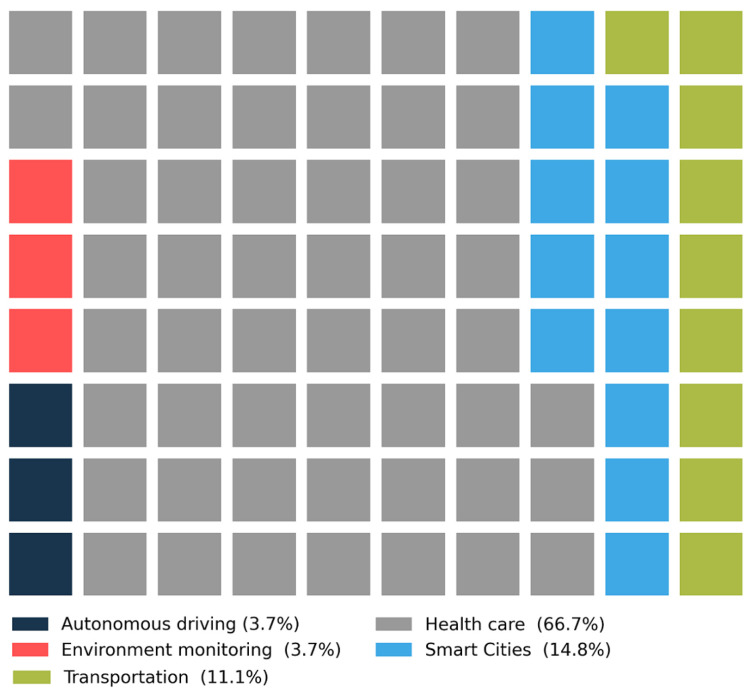
The relationship between HAR and top five most studied application domains.

**Figure 15 sensors-22-05544-f015:**
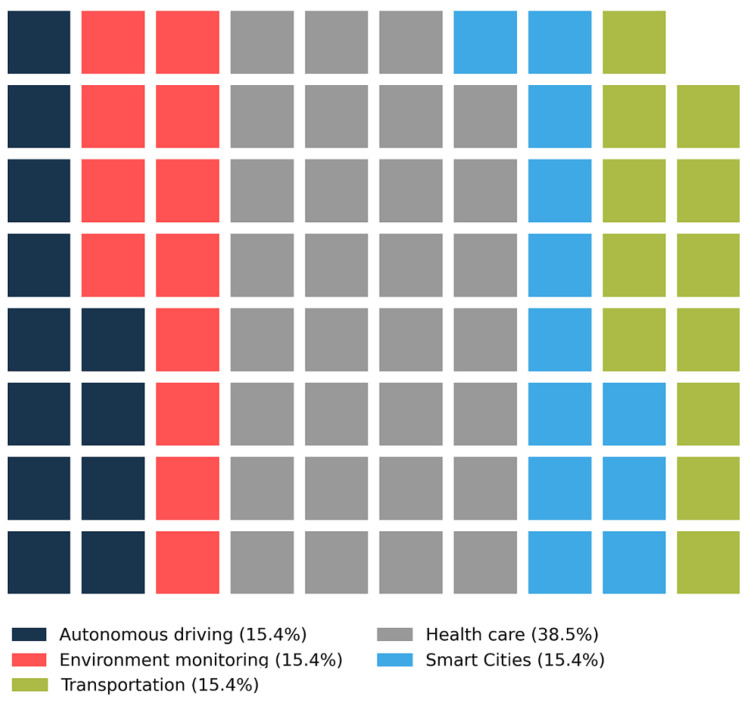
The relationship between QoS and top five most studied application domains.

**Figure 16 sensors-22-05544-f016:**
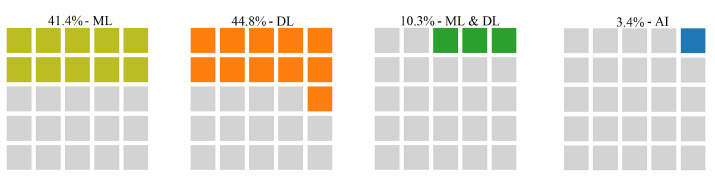
The relationship between HAR challenge and AI techniques categories used to address it.

**Figure 17 sensors-22-05544-f017:**
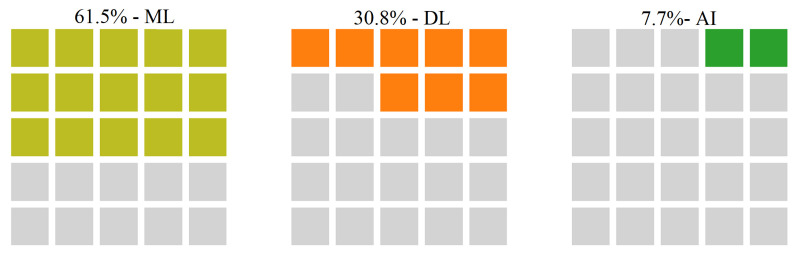
The relationship between QoS challenge and corresponding AI techniques categories used to deal with it.

**Figure 18 sensors-22-05544-f018:**
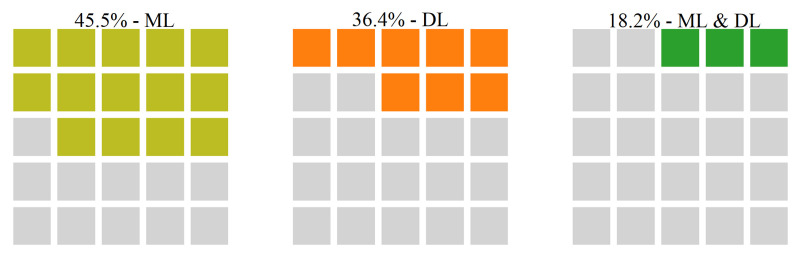
The relationship between Energy Saving challenge and AI techniques categories applied to address it.

**Figure 19 sensors-22-05544-f019:**
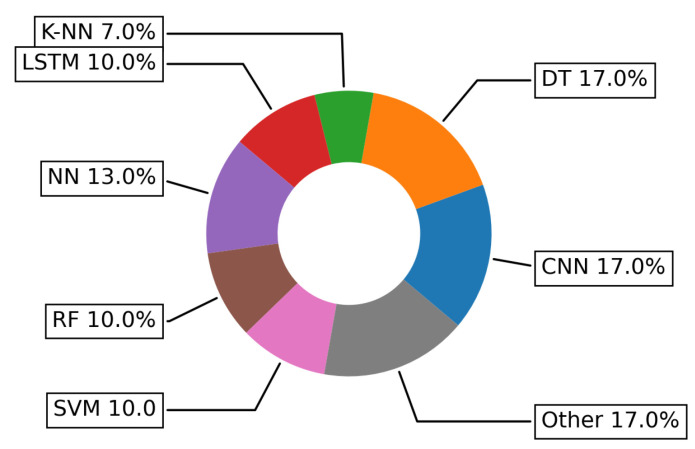
Specific ML and DL algorithms more frequently used in addressing the HAR challenge.

**Figure 20 sensors-22-05544-f020:**
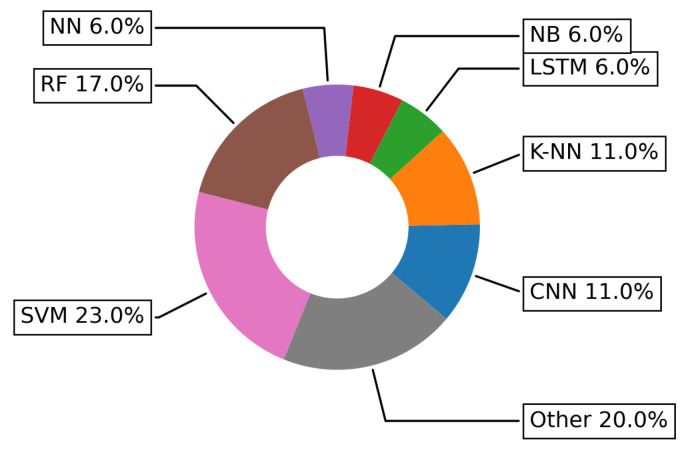
Specific ML and DL algorithms more frequently used in addressing the Monitoring challenge.

**Figure 21 sensors-22-05544-f021:**
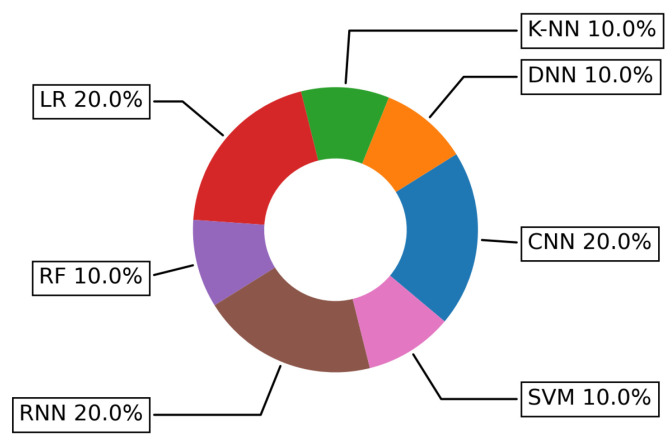
Specific ML and DL algorithms more frequently used in addressing Activity Recognition challenge.

**Figure 22 sensors-22-05544-f022:**
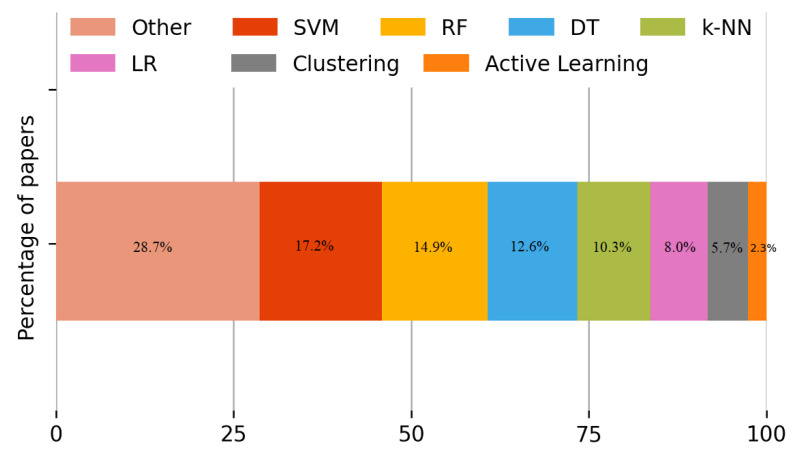
Percentage of papers per ML category of algorithms found in the sample studied. The most used ML techniques are SVM, RF, DT and K-NN.

**Figure 23 sensors-22-05544-f023:**
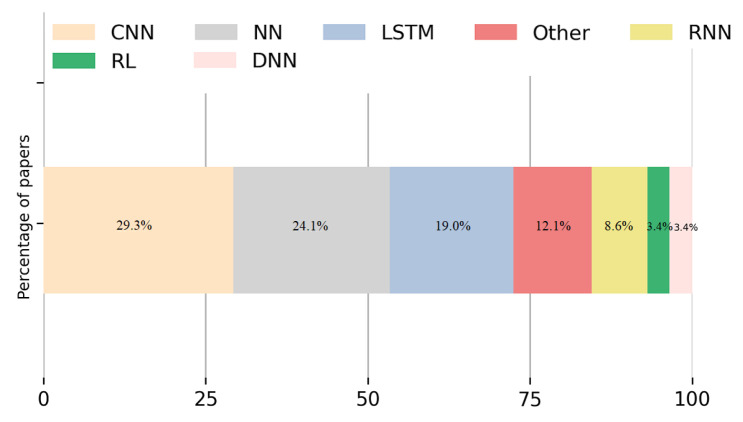
Percentage of papers per DL category of algorithms found in the sample studied. The three most applied DL techniques are CNN, NN and LSTM.

**Figure 24 sensors-22-05544-f024:**
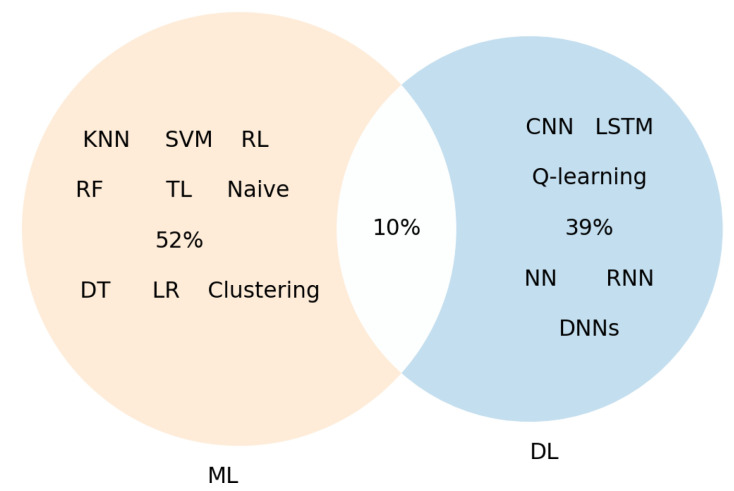
Overview of ML and/or DL techniques that have been used in the included papers.

**Figure 25 sensors-22-05544-f025:**
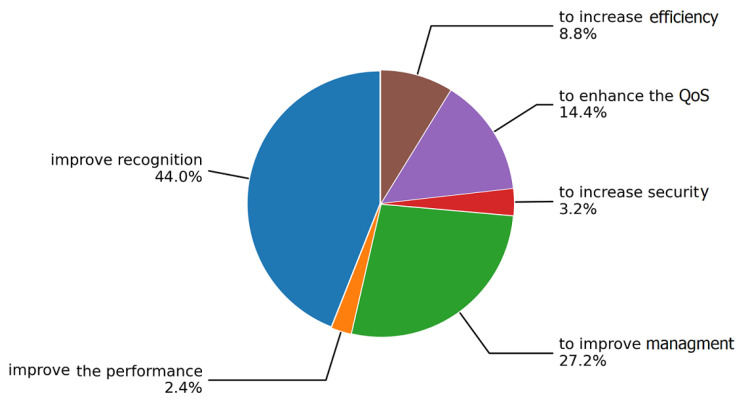
Motivations of adopting AI solutions to context awareness.

**Figure 26 sensors-22-05544-f026:**
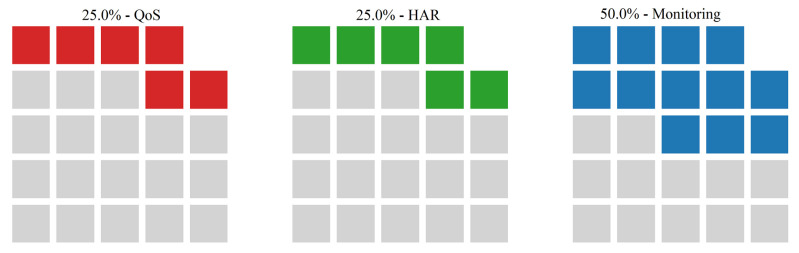
Main challenges in logistics addressed by the papers included in the survey.

**Figure 27 sensors-22-05544-f027:**
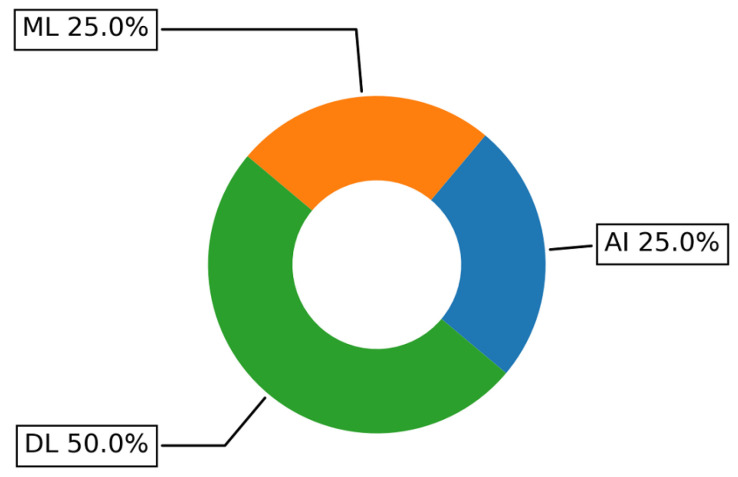
Main AI techniques in logistics addressed by the papers included in the survey.

**Table 1 sensors-22-05544-t001:** Overview of previous related surveys and comparison with our study with respect to their contributions and discussed intelligent techniques (classical AI, ML and DL).

Reference	Year	Main Focus	AI Techniques
[40]	2015	Evaluation of different available resources,	×
		communication mediums, and frameworks	
		for industrial market perspective.	
[41]	2016	A context-aware review for recognizing emerging	×
		fields from a software development	
		point of view.	
[42]	2019	A survey study about context-aware crowd	×
		sensing systems for urban environments.	
[43]	2022	A survey on the use of ML methods in context-	AI, ML and DL
		aware middlewares for HAR.	
[44]	2018	A survey on context awareness for IoT big data analysis.	AI, ML and DL
[45]	2018	A comprehensive survey on the utilization of AI	AI, ML and DL
		integrating ML, data analytics, and NLP techniques	
		for enhancing the efficiency of wireless networks.	
[46]	2019	A literature analysis of various context-aware systems	×
		(modelling, organization, and middleware).	
[47]	2019	A short survey of the latest development	AI, ML and DL
		of context-aware systems.	
[48]	2019	A survey of recent advances in intelligent sensing,	AI, ML and DL
		computation, communication, and energy	
		management for resource-constrained	
		IoT sensor nodes.	
[49]	2021	An extensive survey of AI-based mobile context-	AI, ML and DL
		aware recommender systems.	
Our Paper	2022	A broad study of the adoption	AI, ML and DL
		of edge-based AI solutions for context-	
		awareness in WSNs.	

**Table 2 sensors-22-05544-t002:** Research questions (RQs).

ID	Question
Q1	How much literature activity has there been between 2015 and January 2022?
Q2	What are the challenges in context-aware edge-based AI for sensor networks?
Q3	What are the state-of-the-art solutions used to address the challenges depending on
	the specific application field?
Q4	What are the motivations to adopt AI solutions to context awareness scenario?
Q5	What are the limitations of current literature or what are gaps existing in the current
	research about applying AI technologies to context awareness that future
	researchers can investigate?

**Table 3 sensors-22-05544-t003:** Search strings considering the search process strategy with inclusion and exclusion criteria.

Scientific Database	Search String
Scopus	TITLE-ABS-KEY ((“Context*” OR “aware*”) AND (*edge
	OR device) AND (“artificial intelligence” OR “machine
	learning” OR “deep learning”) AND (“sensor*”))
Web of Science	TS = ((“Context*“ OR “aware*”) AND (*edge OR device)
	AND (“artificial intelligence” OR “machine learning”
	OR “deep learning”) AND (“sensor*”))

**Table 4 sensors-22-05544-t004:** The identified eleven clusters of keywords along with the titles of the subjects (cluster labels) they represent.

Cluster Label	Size	Keywords
AI, ML and DL	27	active learning, AI, ANN, attention mechanism, big data, classification, CNN, data mining, data models, DL, DNN, feature extraction, feature selection, inference, intelligent systems, LSTM, ML, prediction, predictive models, RF, RNN, regression, RL, supervised learning, SVM, time-series classification, training.
Edge Computing and Smart Monitoring	11	EC, pervasive computing, biomedical monitoring, ECG, electrocardiography, health monitoring, heart rate, monitoring, pervasive healthcare, physiological signals, physiology.
Smart Healthcare	10	accelerometer, action recognition, activity recognition, gait recognition, HAR, mhealth, mobile computing, mobile health, mobile sensing, smart healthcare.
Smart and Wearable Devices	8	on-device computation smart devices, smartphone, wearable computing, wearable devices, wearable sensors, wearable system, wearables.
Anticipatory Computing and SSL	4	anticipatory computing, recommendation system, semi-supervised learning, transfer learning.
Context-Awareness	4	context modeling, context-aware systems, context-awareness, context-awareness services.
Energy Consumption and Saving	4	energy consumption, energy efficiency, energy saving, power consumption.
IoT	4	industry 4.0, IoMT, IoT, smart home.
Sensors and WSN	4	WSN, sensor data, sensor fusion, sensors.
Mental Health	3	mental health, stress, stress monitoring.
Computer Vision	3	computer vision, object recognition, pattern recognition.

Note. The clustering is produced by applying DBSCAN clustering with *eps* = 0.3.

**Table 5 sensors-22-05544-t005:** Top ten most frequently used keywords.

Keyword	Occurrences
ML	55
DL	27
IoT	22
activity recognition	17
sensors	12
HAR	10
wearable sensors	10
CNN	8
classification	7
context-awareness	7

**Table 6 sensors-22-05544-t006:** The main subjects along with the references to their related papers.

Main Subject	References	# of Studies
Smart Healthcare	[54,55,56,57,58,59,60,61,62,63,64,65,66,67,68,69,70,71,72,73,74,75,76,77,78,79,80,81,82,83,84,85,86,87,88]	35
AI, ML and DL	[56,59,63,76,78,82,88,89,90,91,92,93,94,95,96,97,98,99,100,101,102,103,104,105,106,107,108,109,110,111,112,113,114]	34
Smart and Wearable Devices	[54,55,56,59,60,61,66,67,78,80,83,86,87,88,89,98,106,109,115,116,117,118,119,120,121,122,123,124,125,126,127]	31
Sensors and WSN	[60,61,76,77,80,83,91,101,104,109,112,113,115,127,128,129,130]	17
Edge Computing and Smart Monitoring	[56,60,67,85,98,106,113,114,117,131,132,133,134,135,136,137]	16
Context-Awareness	[57,65,68,70,76,81,91,93,134,138,139,140,141]	13
Energy Consumption and Saving	[62,63,92,130,137,142,143,144]	8
Anticipatory Computing and SSL	[57,62,68,76,83,119,145]	7
IoT	[77,105,146,147,148,149]	6
Computer Vision	[87,99,150,151,152]	5
Mental Health	[61,80,114,117,136]	5

**Table 7 sensors-22-05544-t007:** Various ML and DL techniques used in context-aware scenarios for sensor networks.

Reference	ML	DL
[60]	SVR, RF, GP, LR, K-NN	ANN
[136]	SVM, J48, RF, NB	NN
[119]	semi-supervised k-means	DNN
[120]	DT, Discriminant Analysis, SVM, K-NN, NB	NN
[176]	Gaussian mixture models	DNN, RNN
[64]	SVM	NN
[65]	RF, DT	NN
[66]	LR	RNN
[121]	RF, SVM, K-NN, SGD, LR, NB, ET	DF
[97]	SVM	NN
[177]	SVM	MLP, LSTM, CNN

## Data Availability

Not applicable.

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
