# Peer review of "Context-Aware Edge-Based AI Models for Wireless Sensor Networks—An Overview"

_sensors, 2022, doi:10.3390/s22155544_

Round 1
Reviewer 1 Report
This paper is well written and addresses an interesting and innovative topic. From my point of view, some discussions could be improved with some more analysis. For example, detecting that location-based services are rare in the literature compared to other applications could be due to a decreased use of the term, as well as for the context-aware term or because these fields are considered solved.. but I would not classify them as gaps or opportunities.
Some minor typos detected in L126, L349 and L351
Another suggestion would be to include the title of the Questions in the structure of the paper in order to have an even better idea of what we are going to find in it.
Congratulations for your work.
Author Response
Response to Reviewer 1 Comments
This paper is well written and addresses an interesting and innovative topic. From my point of view, some discussions could be improved with some more analysis.
Point 1: For example, detecting that location-based services are rare in the literature compared to other applications could be due to a decreased use of the term, as well as for the context-aware term or because these fields are considered solved. but I would not classify them as gaps or opportunities.
Response 1: Our study identify that location-based services are more rarely addressed in the existing studies in the framework of the context-aware wireless sensor networks, not in general. We have clarified this in the manuscript.
Point 2: Some minor typos detected in L126, L349 and L351.
Response 2: Those are corrected.
Point 3: Another suggestion would be to include the title of the Questions in the structure of the paper in order to have an even better idea of what we are going to find in it.
Response 3: According to the reviewer’s suggestion we have added the questions’ titles in the paper structure.

Reviewer 2 Report
An interesting work that includes an extensive bibliometric analysis of AI techniques for context-aware WSN. The work differentiates from other reviews “by applying a semantic-aware approach for identifying the main subjects” [52]. The approach mainly analyzes keywords.
1.- One of the main weakness is the topic overview. Researchers typically expect that a review paper include a description of the technologies, problems to be solved and commonly used approaches on the research field. For example, sections 2 and 3 of reference [49] provide a research field analysis. It seems that this work mainly uses keyword-based analysis and the description of the research areas is limited. For example, section 4.2 selects the “main research topics” (see comment 2) and it includes a brief description of HAR-related research activities and monitoring applications. A reader expects a more detailed description of the research activities that are included in every research topic as well as the types of sensor, contexts and challenges (see comment 2). Additionally, section 4.3 should include more information about the application domains, with less extension than section 5.
2.- It seems that the paper identifies “challenge” (question Q2) with “research topic”. In my option, these concepts are not equivalent because a challenge is a problem to be solved (e.g. section 5.3 motivations) while a research topic is a field of scientific research (e.g. HAR). Therefore, this question should be addressed because it affects paper contributions, question definitions and results.
3.- It seems that section 4.2 “challenges” and section 4.3 “application domains” have been selected after reading the selected papers. The relation between section 4.2 “challenges” and Table 4 research subjects has not clearly described. Moreover, the section 4.4 analysis of subjects (lines 418-441) introduces more confusion. It could be interesting to move lines 418-441 and Table 7 to section 3. Additionally, an analysis of the relations between section 4.2 “research topics/challenges” and the Table 4 subjects could be included on section 4.2.
4.- The comparison with other reviews (section 2.2) should be improved. Table 1 should highlight the paper advantages/improvements; columns AI, ML and DL could be merged in a single column. What are the overview novelties or improvements? Is the use of the semantic-aware approach the main/essential contribution?
5.- The authors points out that “these days, AI, ML and DL are three popular terms that are used interchangeably to characterize systems that behave intelligently”. If the selected papers interchange AI, ML and DL terms, these categories could not be used to classify papers. Therefore, a statement that clarifies the use of these terms in the work is required on section 2.1.3. This is important to understand paper results (e.g. Figures 10 and 23).
6.- The inclusion criteria IC1 excludes papers with the selected terms in the abstract while “Filter 1” selects papers with the selected terms in the abstract. It seems a contradiction.
7.- Tables could improve data analysis and readability on section 4.3. For example, the information of Figures 12-16 could be included in a table that facilities the comparison between “challenges”. A similar approach could be used with Figures 17-19.
8.- The authors comments that the answers to questions Q3 and Q4 are related. In order to provide the relation between application domains, “challenges” and AI techniques, it could be more efficient to merge questions Q3 and Q4 in a single question. Additionally, the first lines of section 4.4 (lines 393-396) should be clarified.
9.- The 1999 definition of “context-awareness” could be updated/improved with reference [23] concepts (e.g. context modelling).
10.- The paper presents an EC system architecture on section 2.1.2. The relation of the proposed architecture and reference [25] types should be commented.
11.- Figure 4 is shocking because it seems that WSN data are delivered directly from the sensor to the users without any edge/fog/cloud computation.
12.- Most of the reviews that evaluate the use of supervised learning techniques do not analyze model training. It could be interesting to include a brief comment about how the selected papers address model training (see for example section 3.5 of https://doi.org/10.1145/3469029).
English style comments:
Line 6: replace an “and” conjunction with a comma.
Line 9: domain in which
Several sentences could be rewritten to improve readability. For example, check lines 100-102, 131-133, 135-138, 176-179 and 255-256.
Author Response
Response to Reviewer 2 Comments
An interesting work that includes an extensive bibliometric analysis of AI techniques for context aware WSN. The work differentiates from other reviews “by applying a semantic-aware approach for identifying the main subjects” [52]. The approach mainly analyzes keywords.
Point 1: One of the main weakness is the topic overview. Researchers typically expect that a review paper include a description of the technologies, problems to be solved and commonly used approaches on the research field. For example, sections 2 and 3 of reference [49] provide a research field analysis. It seems that this work mainly uses keyword-based analysis and the description of the research areas is limited. For example, section 4.2 selects the “main research topics” (see comment 2) and it includes a brief description of HAR-related research activities and monitoring applications. A reader expects a more detailed description of the research activities that are included in every research topic as well as the types of sensor, contexts and challenges (see comment 2). Additionally, section 4.3 should include more information about the application domains, with less extension than section 5.
Response 1: We do not entirely agree with this comment. Our study has been carried to investigate the challenges in context-aware edge-based AI for sensor networks, state-of-the-art solutions used to address these challenges, the motivations to adopt AI solutions to context awareness scenarios, and, finally, the limitations of current literature or what are gaps in the current research about applying AI technologies to context awareness. Furthermore, we have applied a semantic-aware approach solely for identifying the main subjects covered by the 141 papers studied in the survey. The answers of the other research questions are due to a deep study, classification, and analysis of selected papers.
Section 4.2 is devoted to the answer of Q2: "What are the challenges in context-aware edge-based AI for sensor networks?" and does not discuss the main identified survey subjects. The latter ones are presented in Figures 8 and 24 and Table 7, and are discussed in Section 3 and Section 4.4. Regarding the reviewer’s concern shared in the second comment please, see our answer to comment 2.
Sections 4.3 and 4.4 study two types of relations: one between the state-of-the-art solutions used to address the identified main challenges (research problems) and the other between the addressed challenges and corresponding application fields used to evaluate the proposed intelligent solutions. Our study has identified 12 different application domains, but we have focused our analysis only on the top five most studied in the selected papers due to the fact the rest present a very low percentage of the studies, see Figure 20.
The study conducted in reference [49] is discussed and compared to our work in the Related work section.
Point 2: It seems that the paper identifies “challenge” (question Q2) with “research topic”. In my option, these concepts are not equivalent because a challenge is a problem to be solved (e.g. section 5.3 motivations) while a research topic is a field of scientific research (e.g. HAR). Therefore, this question should be addressed because it affects paper contributions, question definitions and results.
Response 2: In our study we have interpreted human activity recognition, continuous monitoring etc. as challenges that have to be solved by applying context-aware AI solutions. We think such consideration of those research problems as challenges would be more useful for the research community. This allows us to identify two important connections, one between the research challenges and application domains and the other one maps the former to the respective intelligent approaches used to solve those. These will eventually reveal the existence of similar challenges (research problems) in different applications, e.g., patient vital sign monitoring or machine health monitoring, and will point out to appropriate intelligent solutions for the problem under consideration that have already been successfully applied to solve the same challenge in other domains.
Point 3: It seems that section 4.2 “challenges” and section 4.3 “application domains” have been selected after reading the selected papers.
The relation between section 4.2 “challenges” and Table 4 research subjects has not clearly described. Moreover, the section 4.4 analysis of subjects (lines 418-441) introduces more confusion. It could be interesting to move lines 418-441 and Table 7 to section 3.
Additionally, an analysis of the relations between section 4.2 “research topics/challenges” and the Table 4 subjects could be included on section 4.2.
Response 3: We think the idea of our semantic-aware approach is not entirely understood by the reviewer. Table 4 depicts the identified eleven clusters of keywords along with the titles of the subjects (cluster labels) they represent while Section 4.2 answers the question Q2 (the main challenges). According to the reviewer's suggestion we have moved the analysis of the identified subjects (Figure 24) and Table 7 to Section 3.
We do not entirely agree with the last statement of the reviewer. Table 4 shows the identified eleven clusters of keywords, while Section 4.2 discusses the main challenges (research problems) that have been identified by analyzing the selected papers. Therefore, we have slightly reworked our manuscript by moving the discussions about the results presented in Tables 4 and 5 and Figure 24 to Section 3 explaining our proposed approach. Then each research question is separately discussed in Section 4.
Point 4: The comparison with other reviews (section 2.2) should be improved. Table 1 should highlight the paper advantages/improvements; columns AI, ML and DL could be merged in a single column. What are the overview novelties or improvements? Is the use of the semantic-aware approach the main/essential contribution?
Response 4: In order to reflect the above remark as well as the comments 3 and 5 of the Editor and comment 2 made by the third reviewer we have expanded the discussion in Section 2.2 “Related work” with further details and comparison with other related studies.
Point 5: The authors points out that “these days, AI, ML and DL are three popular terms that are used interchangeably to characterize systems that behave intelligently”. If the selected papers interchange AI, ML and DL terms, these categories could not be used to classify papers. Therefore, a statement that clarifies the use of these terms in the work is required on section 2.1.3. This is important to understand paper results (e.g. Figures 10 and 23).
Response 5: We have written in the manuscript “these days, AI, ML, and DL are three popular terms that are used interchangeably to characterize systems that behave intelligently” to define the popular use of AI, ML, and DL. The main motivation behind the classification of the extracted papers with respect to the used intelligent methods in three categories (AI, ML and DL) was to get knowledge about the tendency covering the survey period. Namely, it was interesting to notice that it went from more traditional ML techniques, through the boom of the use of DL and in the recent years return to some classical AI modelling methods, e.g., such as fuzzy logic. This is commented in the manuscript.
Point 6: The inclusion criteria IC1 excludes papers with the selected terms in the abstract while “Filter 1” selects papers with the selected terms in the abstract. It seems a contradiction.
Response 6: In IC1, papers with the terms just in the abstract are excluded from this study, because we are looking for these terms in abstract+ title+keywords as a whole. Perhaps, the reviewer has missed this detail.
Point 7: Tables could improve data analysis and readability on section 4.3. For example, the information of Figures 12-16 could be included in a table that facilities the comparison between “challenges”. A similar approach could be used with Figures 17-19.
Response 7: We have a different opinion from the reviewer and believe that the figures better illustrate the identified relationships and are also easier to be understood than tables.
Point 8: The authors comments that the answers to questions Q3 and Q4 are related. In order to provide the relation between application domains, “challenges” and AI techniques, it could be more efficient to merge questions Q3 and Q4 in a single question. Additionally, the first lines of section 4.4 (lines 393-396) should be clarified.
Response 8: According to the reviewer's suggestion Q3 and Q4 are merged and replaced by the following new research question Q3:
"What are the state-of-the-art solutions used to address the challenges depending on the specific application field?" Due to this Sections 4.3 and 4.4 are also merged and slightly reworked.
Point 9: The 1999 definition of “context-awareness” could be updated/improved with reference [23] concepts (e.g. context modelling).
Response 9: The reviewer’s comment is taken into account!
Point 10: The paper presents an EC system architecture on section 2.1.2. The relation of the proposed architecture and reference [25] types should be commented.
Response 10: The reviewer’s comment is taken into account!
Point 11: Figure 4 is shocking because it seems that WSN data are delivered directly from the sensor to the users without any edge/fog/cloud computation.
Response 11: Figure 4 introduces a general WSN architecture. However, in order to reflect the reviewer's comment we have modified the architecture by considering edge/fog/cloud/ computation.
Point 12: Most of the reviews that evaluate the use of supervised learning techniques do not analyze model training. It could be interesting to include a brief comment about how the selected papers address model training (see for example section 3.5 of https://doi.org/10.1145/3469029).
Response 12: It would be interesting to have a discussion on model training in case of supervised learning setup. Unfortunately, most of the extracted studies have not provided specific information about this, even in some papers the used ML algorithms are not specified. Therefore, it would be difficult to get an objective analysis of this aspect. Based on the article mentioned in the reviewer’s comment, where the main focus is on deep learning and inherent problems related to learning with edge devices, we conclude that the reviewer in fact means how the big balanced labelled datasets needed for the training have been ensured in the studied papers. If so this would be an interesting future direction of the current study. It can be devoted to the studying datasets used for the evaluation and validation purposes of the developed intelligent context-aware solutions along with the selected evaluation measures.
English style comments:
Line 6: replace an “and” conjunction with a comma.
Line 9: domain in which
Several sentences could be rewritten to improve readability. For example, check lines 100-102, 131-133, 135-138, 176-179 and 255-256.
All the above suggestions for the readability improvement are taken into account and reflected in the revised version of the manuscript!

Reviewer 3 Report
In this paper, authors review Context-aware edge-based AI models for wireless sensor networks, the topic is very interesting and the paper is well organized and written, However, I have the following suggestions to make the presentation and content of the paper better.
1. Subsections 2.1.3 and 2.1.4 introduce the basics of AI, ML, DL and the WSN. However, these disciplines are basics and can be summarized into short paragraphs in the Introduction section.
2. The limitations of current literature is not analyzed enough. Authors need to elaborate more on the shortcomings of existing works and may need to categorized them based on common issues.
3. The open issues should be also highlighted and presented as points with sufficient discussion and it is better to have separate section or subsection since they are the fruit of this survey.
Author Response
Response to Reviewer 3 Comments
In this paper, authors review Context-aware edge-based AI models for wireless sensor networks, the topic is very interesting and the paper is well organized and written, However, I have the following suggestions to make the presentation and content of the paper better.
Point 1: Subsections 2.1.3 and 2.1.4 introduce the basics of AI, ML, DL and the WSN. However, these disciplines are basics and can be summarized into short paragraphs in the Introduction section.
Response 1: Subsections 2.1.3 and 2.1.4 provide a brief introduction of various concepts used in the study such as AI, ML, DL, edge computing, and the WSN. We prefer to keep them as they are, namely separated, since this will facilitate the reader in understanding the background of the study.
Point 2: The limitations of current literature is not analyzed enough. Authors need to elaborate more on the shortcomings of existing works and may need to categorized them based on common issues.
Response 2: This comment is similar to the comments 3 and 5 of the Editor and comment 4 of the second reviewer therefore, we have expanded the discussion in Section 2.2 “Related work” with further details and comparison with other published reviews.
Point 3: The open issues should be also highlighted and presented as points with sufficient discussion and it is better to have separate section or subsection since they are the fruit of this survey.
Response 3: The open issues are discussed in the last paragraph of the Conclusion section. This discussion is further expanded to include the future research directions.
